# Bioprospecting of Targeted Phenolic Compounds of *Dictyota dichotoma*, *Gongolaria barbata*, *Ericaria amentacea*, *Sargassum hornschuchii* and *Ellisolandia elongata* from the Adriatic Sea Extracted by Two Green Methods

**DOI:** 10.3390/md21020097

**Published:** 2023-01-29

**Authors:** Aly Castillo, Maria Celeiro, Marta Lores, Kristina Grgić, Marija Banožić, Igor Jerković, Stela Jokić

**Affiliations:** 1CRETUS, Department of Analytical Chemistry, Nutrition and Food Science, Universidade de Santiago de Compostela, E-15782 Santiago de Compostela, Spain; 2LIDSA, Departmesnt of Analytical Chemistry, Nutrition and Food Science, Faculty of Chemistry, Universidade de Santiago de Compostela, E-15782 Santiago de Compostela, Spain; 3Department of Process Engineering, Faculty of Food Technology Osijek, University of Josip Juraj Strossmayer in Osijek, Franje Kuhača 18, 31000 Osijek, Croatia; 4Faculty of Agriculture and Food Technology, University of Mostar, Biskupa Čule bb, 88000 Mostar, Bosnia and Herzegovina; 5Department of Organic Chemistry, Faculty of Chemistry and Technology, University of Split, Ruđera Boškovića 35, 21000 Split, Croatia

**Keywords:** Adriatic Sea macroalgae, polyphenols, matrix solid-phase dispersion, ultrasound-assisted extraction, green chemistry

## Abstract

The content of bioactive compounds in four brown and one red algae from the Adriatic Sea (*Dictyota dichotoma*, *Gongolaria barbata, Ericaria amentacea*, *Sargassum hornschuchii* and *Ellisolandia elongata*) is explored. The efficiency of two different extraction methods viz. ultrasound-assisted extraction (UAE) and matrix solid-phase dispersion (MSPD) to obtain the extracts rich in phenolic compounds was compared. The effect of the extraction solvent to modulate the phenolic profile was assessed. In general, the mixture ethanol/water in an isovolumetric proportion showed the best results. The total phenolic content (TPC) and antioxidant activity (AA), as well as the individual polyphenolic profile, were evaluated for five target algae. TPC values ranged between 0.2 mg GAE/g (for *E. elongata*) and 38 mg GAE/g (for *S. hornschuchii*). Regarding the quantification of individual polyphenols by liquid chromatography-tandem mass spectrometry (LC-MS/MS) analysis, the presence of a high number of hydroxybenzoic acid derivatives (mainly of 3- and 4-hydroxybenzoic acids) in all species was noted. In *G. barbata* their concentrations reached up to 500 mg/kg. IC_50_ values (ABTS assay) ranged between 44 mg/L (for *S. hornschuchii*) and 11,040 mg/L (for *E. elongata*). This work contributes to the in-depth characterization of these little-explored algae, showing their potential as a natural source of phenolic compounds.

## 1. Introduction

Phenolic compounds are important for the macroalgae (seaweeds) normal growth and development exhibiting defence mechanisms against infections, injuries, and environmental aggressions [1]. Bioprospecting of macroalgal phenolic compounds has been a challenging issue [1,2,3]. Their structural complexity, along with the high polysaccharide abundance within the macroalgal matrix, make the isolation and characterization of phenolics quite difficult. Usually, a multi-step strategy including different fractionation techniques has been applied, which influenced the effectiveness of their structural characterization [1].

Most studies have reported total phenolic content (TPC) or phlorotannins content (TPhC) only based on spectrophotometric methods, for which the possible co-extraction of, e.g., carbohydrates, may have a significant contribution [1]. Phenolic compounds present in macroalgae vary from simple molecules, such as benzoic acid and cinnamic acid or flavonoids, to the more complex phlorotannin polymeric structures derived from oligomerization of phloroglucinol (1,3,5-trihydroxybenzene) units (PGU) through diaryl ether or C-C bonds, which in some cases are quite exclusive to macroalgae. In addition, some species produce fucophlorethols (especially *Sargassaceae* species), and could produce C-C and diaryl ether bonds [1,4].

For present research, 4 brown macroalgae (*Dictyota dichotoma*, *Gongolaria barbata*, *Ericaria amentacea*, and *Sargassum hornschuchii*) and one red macroalga (*Ellisolandia elongata*) from the Adriatic Sea were selected as part of our continuous research on bioprospecting of the Adriatic Sea (BioProCro). There are limited or scarce data on their chemical composition, especially their polyphenolic content [5]. Higher number of brown algae were chosen, since it is well known that brown macroalgae are rich in phenolics compared to red or green ones which is usually associated with phlorotannins [6,7].

The TPC of *D. dichotoma* from the Adriatic Sea ranged up to ca. 300 mg gallic acid equivalent (GAE)/L, total flavonoid content up to ca. 1000 mg quercetin equivalents (QE)/L, total tannins up to ca. 0.4 mg catechin equivalents (CE)/L and phloroglucinol up to ca. 150 mg/L depending on the extraction solvent [8]. High performance liquid chromatography with ultraviolet detector (HPLC-UV) was used for the analysis of individual phenolic compounds and revealed that the dominant phenolic acids in the extracts of *D. dichotoma* were *trans*-ferulic acid and *p*-coumaric acid followed by protocatechuic acid and *o*-coumaric acid [8]. While this macroalga was investigated from the Adriatic Sea, we re-investigated its phenolic composition and compared it with other investigated macroalgae. The major phenolic acids in the extracts of *D. dichotoma* from the Danish coast identified by HPLC were gentisic acid and protocatechuic acid [9].

TPC determined in *Gongolaria barbata* from Black sea was 385.6 mg GAE/100 g. It contained vanillic acid in the highest quantity followed by benzoic acid and ferulic acid. The smallest quantities were found for gallic acid, pyrogallol acid and 4-aminobenzoic acid [10]. HPLC-UV analysis for phenolic contents of *G. barbata* isolated from the Red Sea coast of Safaga, Egypt revealed (*E*)-vanillic acid, benzoic acid, salicylic acid and ferulic acid as dominant. HPLC-UV analysis of its flavonoids revealed kaempferol-3-glucoside-2-*p*-coumaroyl and rosmarinic acid [11]. Heffernan et al. detected phlorotannins containing up to 16 units of phloroglucinol in *Cystoseira nodicaulis* from the Irish coast [12] while phlorotannins containing up to 17 and 27 PGU were also identified in *Cystoseira abies-marina* [13]. Phlorotannin extracts from *Cystoseira nodicaulis*, *Cystoseira tamariscifolia*, and *Cystoseira usneoides* were characterized [14]. Daily and seasonal variations of optimum quantum yield and phenolic compounds were determined in *Cystoseira tamariscifolia* [2].

Different isoflavones, such as daidzein or genistein, were identified in *Sargassum muticum* and *Sargassum vulgare* [15]. Phlorotannins were identified in *Sargassum wightii* [16]. Phenolic acids [9], flavan-3-ols [17], hydroxytetrafuhalol, triphlorethol and dihydroxypentafuhalol [18] were found in *Sargassum muticum*.

While ultrasound-assisted extraction (UAE) has been applied for the extraction of phenolic compounds from macroalgae by several authors, as evidenced in the comprehensive review by Santos et al. [1], it has never been applied to any of the five algae that are the object of this study. Likewise, the alternative technique that we propose, matrix solid-phase dispersion (MSPD), has not been used in the extraction of bioactive compounds from macroalgae so far. The efficiency and green character of these two extraction techniques for obtaining extracts rich in phenolic compounds are compared. In addition to assessing TPC and the antioxidant activity (AA) in terms of IC_50_, the focus has been placed on the identification and quantification of minor polyphenols by liquid chromatography coupled to tandem mass spectrometry (LC-MS/MS), which is a different approach from the much more investigated phlorotannins.

## 2. Results and Discussion

### 2.1. Solvent Selection

Within the broad group of phytochemical compounds, whether from terrestrial or aquatic plants, phenols are of special interest due to their high bioactive capacity, being important antioxidants associated with beneficial effects on health. While the potential of marine algae for containing these compounds is well known, the studies focused on the extraction and characterization of their polyphenolic profiles, especially for brown algae, are still scarce [19]. As it is well known, phenols present a wide range of polarity and properties depending on their chemical structure. For this reason, the extraction solvent plays a key role in the recovery of these compounds from the algae and, consequently, in the bioactive properties of the obtained extracts [20]. The most used solvent to extract polyphenols from macroalgae, including brown algae, involve methanol, ethanol and their aqueous mixtures and, among them, ethanol is the preferred one for the extracts intended to pharmaceutical, food or cosmetic applications [21].

Therefore, in a first approach, methanol (MeOH) and the hydro-organic mixture ethanol/water (50:50, *v*/*v*) (EtOH-W) were used as the solvents. These preliminary experiments were carried out employing 1 g of the sample extracted by UAE (50 W, 50 °C, 60 min) with 20 mL of the correspondent solvent. The respective TPC results, both in terms of mass/mass and mass/volume, are shown in Figure 1.

As can be seen in Figure 1, TPC values of four brown macroalgae ranged between 7–38 mg GAE/g (341–1800 mg GAE/L) when the mixture EtOH/water was employed as the extraction solvent. In general, by using this hydro-organic mixture, higher TPC values were obtained, while only for *D. dichotoma* the highest TPC value (19% more) was obtained when MeOH was employed as the extraction solvent. Due to the lack of studies dealing with the phenolic content of these brown macroalgae, as well as the different ways of expressing the results in terms of mass or volumetric concentrations, the comparison of the obtained values with others reported in the literature is complex. However, these results are in accordance with several papers for *D. dichotoma*, where TPC values of 19 mg GAE/g and 300 mg GAE/L were reported after using supercritical CO_2_ extraction (SFE) [22] and UAE [21], respectively. However, for *G. barbata* TPC values were close to 25 mg GAE/g, while lower TPC values (3.9 mg GAE/g) were reported in the literature [10].

For red macroalga *E. elongata*, although its capacity to contain carotenoid compounds as a photoprotective arsenal is well known, with zeaxanthin standing out [23], its potential as a supplier of phenolic compounds was so far unknown. Present research determined the lowest TPC values (0.6 mg GAE/g, 29 mg GAE/L) among all samples. The lower concentration of phlorotannins in red algae may also contribute to the lower TPC values [1]. Similarly, the bioactive capacity of *E. amentacea*, not only based on the potential phenolic content, but on other phytochemicals, is still unknown. In the present research, this alga showed TPC of 8.6 mg GAE/g. It is important to highlight that similar or even higher TPC values for these two species were observed in comparison to other brown algae that are far more studied and better characterized, such as *Sargassum fusiforme* or *Halopteris scoparia* [21].

As can be observed in Figure 1, the effect of the solvent plays key role in the extraction of all the algae studied, mainly for *C. barbata*, showing TPC values ten times higher when the mixture EtOH/water (50:50, *v*/*v*) is employed as the extraction solvent in comparison to methanol. This behaviour was previously observed by other authors, where the aqueous extraction of this algae showed phenolic content 10.6 times higher than the methanolic one [24]. However, for the species with the highest TPC, such as *S. hornschuchii*, the use of different solvent exhibited the lowest impact among the analyzed macroalgae. The obtained value for this specie of 38 mg GAE/g (using EtOH/water) is in accordance with those reported in the literature for other *Sargassum* species such as *S. confusum* (21 mg GAE/g) [21].

### 2.2. Comparison between UAE and MSPD to Extract Polyphenols

Based on the previous results, the mixture ethanol/water (50:50, *v*/*v*) was selected as the extraction solvent for further experiments, as it enables a clear improvement in the overall TPC values. Besides, ethanol is a Generally Recognised As Safe (GRAS) solvent according to the United States Food and Drug Administration (FDA) and its use in food products is allowed by the European Food Safety Authority (EFSA) [25].

One important point to be considered in the recovery of bioactive compounds from the algae is the extraction procedure. As was commented, UAE has been the most widely applied to this matrix, showing different advantages such as the possibility of performing simultaneous extractions [26,27,28]. However, its main drawback is that the effect of the ultrasound waves cannot be homogeneous during the extraction time, and several authors have reported that although the use of high temperatures usually leads to a kinetic improvement, most phenolics are not thermostable, so the heat treatments could reduce their total extracted amount [8,29]. Besides, centrifugation of UAE derived extracts is usually required to separate the extract containing the compounds of interest from the algae residue, increasing the number of steps necessary for the sample preparation, and resulting in a higher extraction time and energy consumption [30]. Due to the complexity of the sample matrix, a technique capable of breaking the algae cells to release the bioactive compounds is necessary [31,32]. In this way, the use of MSPD results in a very suitable tool [33]. Both experimental procedures are described in Section 3.3 and depicted in Figure 2.

In the last years, the development and use of extraction procedures fulfilling the green analytical chemistry (GAC) and green sample preparation (GSP) principles is increasing [34,35]. They include the use of safe solvents/reagents and materials, minimizing waste generation and energy demand, and enabling high sample throughput to minimize the negative environmental impact. In addition, it is also important to evaluate the interconnections between different experimental parameters involved in the extraction step. Very recently, the AGREEPrep metric tool [36] was developed for assessing the greenness of sample preparation methods. It is based on the principles of GSP. The greenness of both UAE and MSPD under the employed experimental conditions was assessed and results are shown in Figure 3. The values closer to one indicate a higher degree of greenness. Values of 0.46 and 0.72 were obtained for UAE and MSPD, respectively.

As can be seen, main differences are observed for item 4 (waste generation), item 7 (number of steps) and item 8 (energy consumption). In comparison with UAE, MSPD combines three procedures in a single step: (i) Disruption of the sample and dispersion onto a solid-phase support, (ii) maceration (close contact between the sample and extraction solvent), and (iii) in-situ extract clean-up, with zero energy consumption (extraction is performed at ambient temperature and pressure) and zero-waste generation, fulfilling with the GAC and GSP principles. Therefore, from greenness point of view, the use of MSPD is more environmentally friendly than UAE. This technique was successfully employed for the extraction of carotenoids [32] and fat-soluble vitamins [37] from microalgae, as well as in combination with vortex-assisted extraction (VAE) to determine halogens in edible seaweed [38], but to the best of our knowledge, it has never been applied to extract the bioactive compounds from macroalgae. Thus, both UAE and MSPD procedures were compared in terms of efficiency.

#### 2.2.1. TPC and AA Analysis

After UAE and MSPD procedures, all extracts were analyzed by spectrophotometry to obtain TPC and AA values. The values are shown for four sequential 5 mL extractions to determine the extractive behaviour of phenols in successive extractions. The results are summarized in Figure 4 and individual values for each algae and extract fraction are shown in Table 1.

#### 2.2.2. IC_50_ Results

The IC_50_ value, a parameter widely employed to measure the antioxidant activity of bioactive extracts, was also assessed and the results are summarized in Figure 5a for the first aliquot of 5 mL obtained by UAE and MSPD procedures.

The IC_50_ value was calculated as the concentration of antioxidants needed to decrease the initial ABTS concentration by 50% (experimental details are shown in Section 3.5). Thus, lower IC_50_ values represent higher antioxidant activity. As can be seen in Figure 5a, IC_50_ values ranged between 44 mg/L (for *S. hornschuchii*) and 11,040 mg/L (for *E. enlongata*). As was previously commented, there is a lack of information in the literature regarding the characterization of these algae. However, the obtained IC_50_ values are below those reported by other authors for brown macroalgae, *Sargassum binderi* (36,627 mg/L), and *Turbinaria conoides* (96,242 mg/L) from the Gulf of Thailand [39,40], employing water as the extractant. The obtained IC_50_ values in this work are also lower than those reported for *Sargassum muticum* (450–520 mg/L) from the Cost of Brittani (France) after employing classical solid-liquid extraction and the advanced technique centrifugal partition extraction (CPE) using ethanol/water (50:50, *v*/*v*). These results demonstrate the great antioxidant potential of the extracts obtained from these algae from the Adriatic Sea. Figure 5b,c show the corresponding TPC and AA values, respectively, for the first 5 mL aliquots obtained by MSPD and UAE. The direct correlation among them is clear (Figure 5b,c), indicating that polyphenols (mainly phlorotannins) are the key compounds that reduce oxidation and being responsible for ABTS scavenging capability of the algae extracts.

#### 2.2.3. Quantification of Individual Polyphenols

Polyphenols are one of the most common classes of secondary metabolites found in terrestrial plants as well as in marine algae. However, there are fundamental differences in the chemical structures of polyphenols in both terrestrial and marine plants. Polyphenols from terrestrial plants are mainly derived from gallic and ellagic acids, whereas the marine algae polyphenols are derived from phloroglucinol [41]. Marine algae polyphenols are known as phlorotannins. They possess a high hydrophilic character and are formed by polymerized phloroglucinol units [42,43]. In general, polyphenols have been little or not at all explored in marine algae and the identification and characterization of polyphenols in target algae has not been sufficiently studied. In fact, to the best of our knowledge, the individual polyphenolic profile has not been evaluated for *E. amentacea*, and *G. barbata*. In this work, 60 target phenols (see Appendix A) were analyzed by LC-MS/MS analysis. Fifteen of them were identified and quantified in the extracts. The results are summarized in Table 2 and Figure 6.

As can be seen in Table 2 and Figure 6, both flavonoids and non-flavonols polyphenols were detected. High number of compounds were detected in *D. dichotoma* and *G. barbata*, whereas for the other three species, a low number of compounds were quantified. Non flavonoids (derivatives of hydroxybenzoic acid (salicylic acid)) were detected in all analyzed algae. Salicylic acid is one of the oldest compounds used in medicine, either free or in the ester form [44]. Its antipyretic, antirheumatic and analgesic properties have been largely demonstrated and, recent studies showed not only its antioxidant activity, but also the activity of its derivatives [45]. In the obtained *G. barbata* extracts the presence of 3-hydroxybenzoic acid and 4-hydroxybenzoic acid at the concentration levels up to 500 mg/kg was determined. These results are in accordance with those observed in ethanolic extracts of Tunisian *G. barbata*, were high amount (76.8 mg/L) of salicylic acid was also found [11]. In recent study related to the antioxidant and antimicrobial activities of naturally occurring compounds in Tunisian *G. barbata*, in addition to the already known major phlorotannins usually found in algae (phloroglucinol and their derivatives), several flavonoids were identified including quercetin and its derivatives, although they were not quantified [46]. In present work, quercetin dominates among flavonoids and, it was found in all species at concentrations ranging from 1.8 to 11 mg/kg. 3- and 4-hydroxybenzoic acids show antifungal and antimicrobial activities [47,48]. While there is a lack of studies regarding the marine algae in comparison with terrestrial plant flavonoids, recent studies suggest that flavonoid enriched functional foods are able to prevent neurodegenerative disorders and heart diseases [49]. Several of the compounds detected in the targeted algae, such as quercetin, exhibited a high antioxidant character that is related to its chemical structure (the position of -OH hydroxyl group [50]). In addition, its antibacterial activity against *S.aureus* and *E. coli* has been demonstrated [51].

In the comparison between UAE and MSPD, for the more polar compounds such as 3,4-, 2-4- and 2-5-dihydroxybenzoic acids, the extraction yield of MSPD is considerably higher than UAE. It is important to note that the extraction solvent was EtOH/water in isovolumetric proportion. These compounds possess a higher affinity for water, and it seems that extraction under mild conditions without temperature (MSPD) favours the transfer of the analytes to water. In addition, the use of UAE (higher temperature) may favour the loss or decomposition of these compounds. However, the differences observed for quercetin between UAE and MSPD may be related to the fact that quercetin-glycosides are easily decomposed [52]. As can be seen in Table 2, the concentrations for quercetin derivatives are very similar using both techniques, while the concentrations for quercetin are considerably higher after UAE.

## 3. Materials and Methods

### 3.1. Sampling

*Dictyota dichotoma* (Hudson) J.L. Lamouroux, 1809 was collected in February 2022 close to Zadar, Adriatic Sea, Croatia (44 05 50 N; 15 14 43 E) from depth of 1 m. *Gongolaria barbata* (Stackhouse) Kuntze (ex. *Cystoseria barbata*) was collected in May 2021 close to Zadar, Adriatic Sea, Croatia (44 12 42 N; 15 09 23 E) at depth of 4–6 m. *Ericaria amentacea* (C. Aghard) Molinary and Guiry 2020 (ex *Cystoseria spicata*) was gathered in April 2021 at Dugi otok, Adriatic Sea, Croatia (44 03 16 N; 14 59 14 E) from depth of 0.5 m. *Sargassum hornschuchii* C. Agardh, 1820 was collected in December 2021 at Mandre, island Pag, Adriatic Sea, Croatia (44 28 37 N; 14 54 58 E) from depth of 20 m. *Ellisolandia elongata* (J.Ellis and Solander) K.R.Hind & G.W.Saunders, 2013 was collected in November 2021 at Dugi otok, Adriatic Sea, Croatia (44 03 16 N; 14 59 14 E) from depth of 0.5 m.

Fresh samples of macroalgae were washed separately five times in water and twice in deionized water. For the freeze-drying experiment the samples were cut in a slice and firstly frozen at −60 °C for 24 h in an ultra-low freezer and then placed in a laboratory freeze dryer (CoolSafe PRO, Labogene, Denmark). The freeze drying process was performed for 24 h under high vacuum (0.13–0.55 hPa) with primary and secondary drying temperatures of −30 °C and 20 °C, respectively. Freeze-dried macroalgae were further used for the extraction.

### 3.2. Standards, Reagents and Materials

For 60 target polyphenols, their CAS numbers, molecular mass, retention time and MS/MS transitions are shown in Appendix A. Water and methanol (MS grade), and ethanol were supplied by Scharlab (Barcelona, Spain). The Folin-Ciocalteu’s phenol reagent (2M), 6-hydroxy-2,5,7,8-tetramethylchroman-2-carboxylic acid (Trolox^®^), 2,2′-azino-bis(3-ethylbenzothiazoline-6-sulfonic acid) diammonium salt (ABTS), formic acid, sodium carbonate and sand (200–300 µm mesh) were supplied by Sigma-Aldrich (Darmstadt, Germany).

### 3.3. Extraction Procedures: UAE and MSPD

#### 3.3.1. Ultrasound-Assisted Extraction (UAE)

1 g of freeze-dried sample was placed in a 10 mL glass vial and 5 mL of the correspondent solvent (methanol or ethanol/water (50:50, *v*/*v*)) were added. Then, the vial was encapsulated and immersed in an ultrasonic bath (J.P. Selecta, Barcelona, Spain) for 60 min at 50 °C. After the extraction, the extract was collected, centrifuged (Orto Alresa, Barcelona, Spain) at 3500 rpm for 10 min and the precipitate was re-extracted with 5 mL of fresh solvent (the procedure was repeated 4 times). Afterwards, the extract was filtered through 0.22 µm polytetrafluoroethylene (PTFE) filters before the analysis.

#### 3.3.2. Matrix Solid-Phase Dispersion (MSPD)

1 g of freeze-dried sample was gently blended with 8 g of dispersing sorbent (sand) into porcelain mortar using a porcelain pestle until a homogeneous mixture was obtained. The mixture was transferred into a polypropylene cartridge containing a PTFE cellulose frit at the bottom, and 0.5 g of sand (to obtain a further degree of fractionation and sample clean-up). Finally, other cellulose frit was placed at the top to compress the mixture. Elution was made by gravity flow with ethanol/water (50:50, *v*/*v*), subsequently collecting 5 mL (×4) of the extract into different volumetric flasks.

### 3.4. Total Polyphenolic Content (TPC)

TPC was determined by employing the Folin-Ciocalteu (FC) colorimetric method described by Singleton and Rossi [53] employing a modification of the Zhang’s guidelines [54] for microtitration in 96-well plates. Briefly, 20 µL of each diluted extract was mixed with 100 µL of Folin-Ciocalteu reagent (1:10, *v*/*v*) and 80 µL of sodium carbonate aqueous solution (7.5 g/L). The mixture was shaken and kept in the dark for 30 min. Afterwards, the absorbance was measured at 760 nm in a microplate reader (BMG LABTECH, Ortenberg, Germany). The TPC was quantified by employing a calibration curve of gallic acid covering a concentration range between 20 and 160 mg/L (absorbance: 0.200–0.800). The TPC was expressed as milligrams of gallic acid equivalent per liter of extract (mg GAE/L) and per gram of the sample (dry weight) (mg GAE/g).

### 3.5. Antioxidant Activity (AA) and IC_50_

AA was determined using the ABTS reagent following the method described by Ozgen et al. [55]. Briefly, 100 µL of the extracts at different dilutions factors were placed in a 96-well plate and mixed with 100 µL of ABTS reagent (prepared in methanol). The mixture was kept in the dark for 10 min and the measurement was performed at 515 nm in a microplate reader (BMG LABTECH, Ortenberg, Germany). For the quantification of AA, a calibration curve prepared in Trolox, covering a concentration range between 3 and 31 mg/L (absorbance between 0.200–0.800), was employed. The AA was expressed as millimoles Trolox equivalent per liter of the extract (mmol TRE/L).

The mean inhibitory concentration (IC_50_) of the samples was also calculated, represented as the milligrams of dry alga per litre of extract (mg·L^−1^) necessary to inhibit 50% of the ABTS radicals [56]. For this purpose, 8 concentration levels of the extract were carried out in a range of inhibition between 20% and 80% of ABTS^+^. In this way, it interpolates in the linear range close to 50% by means of the relationship
Y=aX+b
where:

“*Y*” is the percentage inhibition of ABTS+.

“*X*” the concentration of the extract in mg/L

“*a*” and “*b*” are the fitted parameters of the regression line.

### 3.6. Liquid Chromatography-Tandem Mass Spectrometry (LC-MS/MS)

For the quantification of the targeted polyphenols in the algae extracts, LC-MS/MS analysis was performed by employing a Thermo Scientific (San José, CA, USA) instrument based on a TSQ Quantum Ultra^TM^ triple quadrupole mass spectrometer equipped with a HESI-II (heated electrospray ionization) source and an Accela Open autosampler with a 20 μL loop. The chromatographic separation was performed by using a Kinetex C18 column (2.6 μm, 100 × 2.1 mm) with a guard column (SecurityGuard^TM^ ULTRA Holder) obtained from Phenomenex (Torrance, CA, USA). The column temperature was set at 50 °C, and the injection volume was 10 μL. The mobile phase consisted of water (A) and methanol (B), both containing 0.1% formic acid. The eluted program started with 5% of B (held 5 min), and it was up to 90% of B over 11 min (held 3 min). Then, initial conditions were reached in 5 min. The flow rate was kept constant at 200 μL/min. The total run time for each injection was 20 min. The mass spectrometer and the HESI-II source were working simultaneously in the positive and negative mode. Selected Reaction Monitoring (SRM) acquisition mode was implemented monitoring 2 or 3 transitions per compound (see Appendix A), for an unequivocal identification and quantification of the target compounds. The system was operated by Xcalibur 2.2 and Trace Finder^TM^ 3.2.

## 4. Conclusions

This work provides useful knowledge on the phenolic content and radical scavenging properties of four brown algae *Dictyota dichotoma*, *Gongolaria barbata*, *Ericaria amentacea*, *Sargassum hornschuchii* and 1 red algae *Ellisolandia elongata* from the Adriatic Sea. The ethanolic/water extracts showed high TPC and a good antioxidant potential. A comparison between two extraction techniques, MSPD and UAE, was made, showing their suitability in terms of efficiency; although according to their greenness character, MSPD offers a low-cost and more environmentally-friendly alternative to UAE, demonstrating that 79% of the polyphenols are extracted in only 5 mL of extract. TPC values ranged between 0.3–43 mg GAE/g and AA between 0.1–21 mmolTRE/100g. Fifteen of 60 polyphenols analyzed by LC-MS/MS were detected in the samples, including both flavonoids and non flavonoids, highlighting the presence of 3- and 4-hydroxybenzoic acids at concentrations up to 543 mg/kg in *G. barbata*. IC_50_ values were also assessed, ranging between 44 mg/L (for *S. hornschuchii*) and 11,040 mg/L (for *E. elongata*). This is a little-explored area and further research should be performed to obtain a deep characterization of these promising algae as a source of bioactive compounds for their re-valorization and potential use in the food, cosmetic and/or pharmaceutical industries. This work contributes to the in-depth characterization of these little-explored algae, showing their potential as a natural source of phenolic compounds.

## Figures and Tables

**Figure 1 marinedrugs-21-00097-f001:**
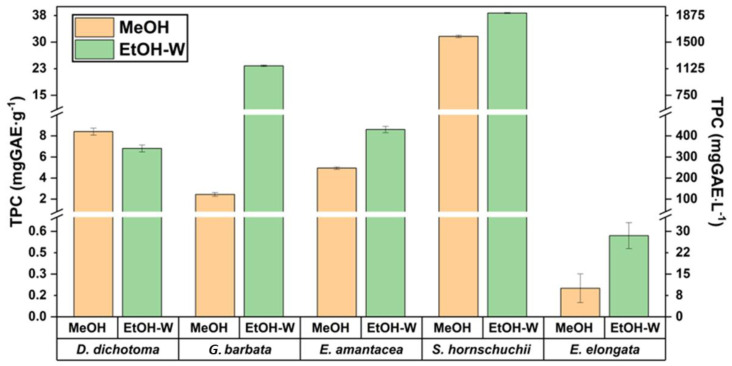
TPC values for the five analyzed algae. MeOH (methanol), EtOH-W (ethanol/water, 50:50, *v*/*v*).

**Figure 2 marinedrugs-21-00097-f002:**
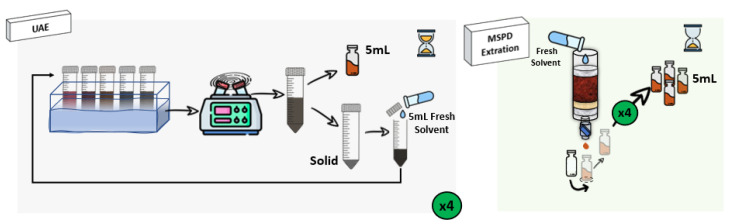
Outline of UAE and MSPD extraction procedures to extract bioactive compounds from the target macroalgae.

**Figure 3 marinedrugs-21-00097-f003:**
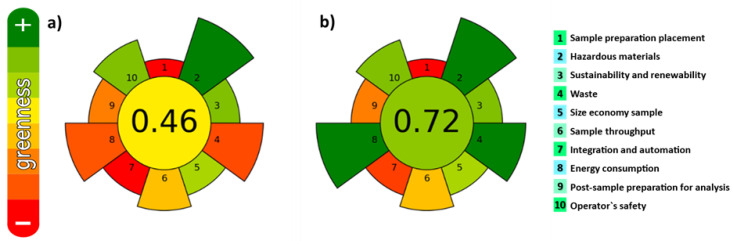
Evaluation of the degree of greenness. Pictograms obtained for (**a**) UAE; (**b**) MSPD.

**Figure 4 marinedrugs-21-00097-f004:**
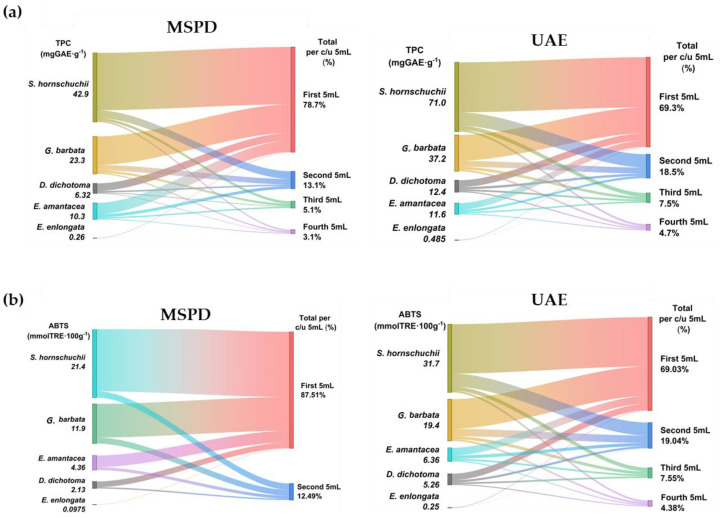
TPC (**a**) and AA values (**b**) for the extracts obtained by MSPD and UAE.

**Figure 5 marinedrugs-21-00097-f005:**
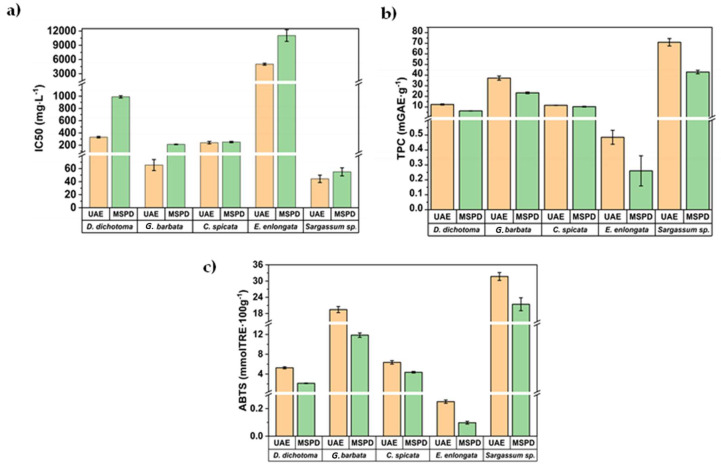
Comparison of the (**a**) IC_50_, (**b**) TPC, and (**c**) AA values for the first aliquot obtained by MSPD and UAE.

**Figure 6 marinedrugs-21-00097-f006:**
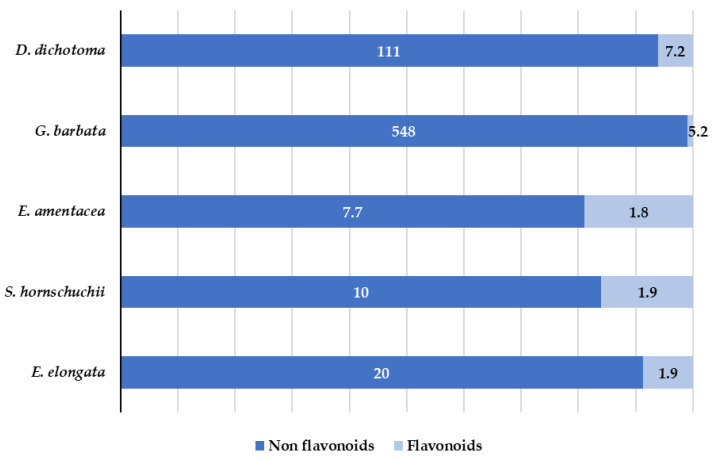
Distribution (mg/kg) of flavonoids and non-flavonoids in the targeted algae.

**Table 1 marinedrugs-21-00097-t001:** TPC and AA values for the four sequential 5 mL extractions for each algae extract by MSPD and UAE.

Sample	Collected Aliquot	TPC (mgGAE/g)		TPC (mgGAE/L)		ABTS (mmolTRE/100g)		ABTS (mmolTRE/L)		IC_50_ (mg/L)	
		MSPD	UAE	MSPD	UAE	MSPD	UAE	MSPD	UAE	MSPD	UAE
** *D. dichotoma* **	1st 5 mL	4.7	7.0	938	1414	1.7	3.0	3.4	6.0	989	331
	2nd 5 mL	0.8	2.3	177	470	0.4	1.0	0.8	2.00	5310	796
	3rd 5 mL	0.4	1.8	87	364	<LDL	0.7	<LDL	1.56	>UDL	1279
	4th 5 mL	0.3	1.1	61	225	<LDL	0.4	<LDL	0.90	>UDL	1928
** *G. barbata* **	1st 5 mL	17	26	3546	5294	10	13	20	27.54	213	65
	2nd 5 mL	3.3	6.1	678	1231	1.7	3.4	3.6	6.85	1189	288
	3rd 5 mL	1.3	2.6	268	539	<LDL	1.3	<LDL	2.73	>UDL	733
	4th 5 mL	0.8	1.9	168	381	<LDL	0.8	<LDL	1.76	>UDL	1014
** *E. amentacea* **	1st 5 mL	7.1	6.8	1421	1361	3.4	3.5	6.9	7.13	251	240
	2nd 5 mL	1.8	2.2	361	444	0.8	1.4	1.77	2.91	955	689
	3rd 5 mL	0.8	1.4	173	294	<LDL	0.8	<LDL	1.61	>UDL	1212
	4th 5 mL	0.5	1.0	103	212	<LDL	0.5	<LDL	1.07	>UDL	1552
** *S. hornschuchii* **	1st 5 mL	38	51	7128	10240	19	23	59	46	55	44
	2nd 5 mL	4.7	13	954	2725	1.9	6.0	3.8	12	434	125
	3rd 5 mL	1.6	4.0	324	810	<LDL	1.8	<LDL	3.6	>UDL	470
	4th 5 mL	0.8	2.0	176	417	<LDL	0.9	<LDL	1.7	>UDL	951
** *E. enlongata* **	1st 5 mL	0.2	0.3	42	74	0.1	0.2	0.2	1.2	11040	5006
	2nd 5 mL	0.02	0.1	<LDL	23	<LDL	0.05	<LDL	0.26	>UDL	18,657
	3rd 5 mL	0.01	<LDL	<LDL	<LDL	<LDL	<LDL	<LDL	<LDL	>UDL	>UDL
	4th 5 mL	0.02	<LDL	<LDL	<LDL	<LDL	<LDL	<LDL	<LDL	>UDL	>UDL

LDL: Lower detection limit, UDL: Upper detection limit.

**Table 2 marinedrugs-21-00097-t002:** Concentration, expressed as mg/kg (dry weight), of the polyphenols detected in the analyzed species by UAE and MSPD.

Polyphenols	*D. dichotoma*		*G. barbata*		*E. amentacea*		*S. hornschuchii*		*E. elongata*	
	*UAE*	*MSPD*	*UAE*	*MSPD*	*UAE*	*MSPD*	*UAE*	*MSPD*	*UAE*	*MSPD*
** *Non flavonoids* **										
3,4-dihydroxybenzoic acid	14 ± 1	40 ± 10	0.42 ± 0.01	1.3 ± 0.4	nd	nd	0.24 ± 0.02	1.6 ± 0.2	nd	nd
2,4-dihydroxybenzoic acid + 2-5-dihydroxybenzoic acid	9.3 ± 0.4	26 ± 4	0.29 ± 0.03	1.0 ± 0.1	0.05 ± 0.01	0.4 ± 0.1	0.11 ± 0.01	0.8 ± 0.1	nd	nd
2,5-dihydroxybenzaldehyde + 3,4-dihydroxybenzaldehyde	0.16 ± 0.01	0.20 ± 0.02	0.17 ± 0.03	0.12 ± 0.02	0.14 ± 0.06	0.21 ± 0.04	nd	nd	nd	nd
3-hydroxybenzoic acid + 4-hydroxybenzoic acid	12 ± 3	13 ± 5	393 ± 19	543 ± 75	8 ± 1	6 ± 2	6 ± 1	8 ± 3	15 ± 1	16 ± 1
2,6-dihydroxybenzoic acid + 3-5-dihydroxybenzoic acid	11 ± 1	31 ± 5	nd	1.2 ± 0.1	nd	nd	nd	nd	nd	nd
3-hydroxybenzaldehyde + 4-hydroxybenzaldehyde	1.8 ± 0.2	0.9 ± 0.2	1.1 ± 0.2	1.0 ± 0.2	2.1 ± 0.1	1.1 ± 0.3	nd	nd	6 ± 1	4 ± 1
** *Flavonoids* **										
Quercetin-3-glucuronide	0.07 ± 0.01	0.3 ± 0.1	0.12 ± 0.02	0.3 ± 0.1	nd	nd	nd	nd	nd	nd
Quercetin-3-glucoside	0.12 ± 0.01	0.08 ± 0.01	0.09 ± 0.03	0.14 ± 0.01	nd	nd	nd	nd	nd	nd
3,5-dimethoxybenzaldehyde	0.10 ± 0.03	2.5 ± 0.9	nd	1.8 ± 0.6	nd	nd	nd	nd	nd	nd
Quercetin	10 ± 1	4.3 ± 0.4	11 ± 3	3 ± 1	3 ± 1	1.8 ± 0.8	9 ± 1	1.9 ± 0.3	15 ± 2	1.9 ± 0.6

nd: Not detected.

## Data Availability

Not applicable.

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
