# Peer review of "Bioprospecting of Targeted Phenolic Compounds of Dictyota dichotoma, Gongolaria barbata, Ericaria amentacea, Sargassum hornschuchii and Ellisolandia elongata from the Adriatic Sea Extracted by Two Green Methods"

_marinedrugs, 2023, doi:10.3390/md21020097_

Round 1

Reviewer 1 Report

The authors have done a wonderful job of valorizing macroalgae of industrial importance. I congratulate the authors for their work and contribution. 

Reviewer Comments:

1.       In line 36, the term ‘Polyphenols’ was repeated in keywords.

2.       In line 93, change ‘solid phase’ into ‘solid-phase’.

3.       Line 204, ‘in-situ’ change it into italics.

4.       Check the lines from 298 to 301.

5.       Full-stop is missing in line number 416.

Author Response

Comments to the Author: The authors have done a wonderful job of valorizing macroalgae of industrial importance. I congratulate the authors for their work and contribution. 

Reply. We appreciate this reviewer’s comment very much.

  1. In line 36, the term ‘Polyphenols’ was repeated in keywords.

Reply. Revised as suggested.

  1. In line 93, change ‘solid phase’ into ‘solid-phase’.

Reply. Revised as suggested.

  1. Line 204, ‘in-situ’ change it into italics.

Reply. Revised as suggested.

  1. Check the lines from 298 to 301.

Reply. It is written more clearly.

  1. Full-stop is missing in line number 416.

Reply. Revised as suggested.

Reviewer 2 Report

The authors have presented the research in a very poor manner. The units have not been expressed properly. Many experiments were found wrongly represented. Same data with table and figures are not required. Needs elaborate revision before acceptance

Author Response

Comments to the Author: The authors have presented the research in a very poor manner. The units have not been expressed properly. Many experiments were found wrongly represented. Same data with table and figures are not required. Needs elaborate revision before acceptance.

Reply. Thank you for the remarks and constructive suggestions. We adopted the corrections and provided replies to Your questions and remarks which we beleived that improved the quality of the manuscript.

  1. Author citation is needed

Reply. This will be added by editorial staff during production.

  1. In Abstract section, line 23, “is detail explored”, Sense of the sentence is not understandable.

Reply. It is revised.

  1. “The efficiency of two different extraction methods (ultrasound-assisted extraction (UAE) and matrix solid-phase dispersion (MSPD) to obtain the extracts rich in phenolic compounds was compared.”………. omit first bracket before “ultrasound” and also mention “viz.” before the names of the two methods of extraction.

 Reply. Revised as suggested.

  1. Line 26 and 27 omit the hyphenated words like re-sults and pol-yphenolic and in throughout the whole manuscript

Reply. Revised as suggested.

  1. “Regarding the quantification of individual 29 polyphenols by liquid chromatography-tandem mass spectrometry (LC-MS/MS) analysis, the pres- ence of a high number of hydroxybenzoic acid derivatives (mainly of 3- and 4-hydroxybenzoic ac- ids) in all species was noted”…………. Split the sentence into two understandable sentences

Reply. Revised as suggested.

  1. Line 42, replace “macroalgae” with “macroalgal” and in the whole text

Reply. Revised as suggested.

  1. Too many grammatical error.

Reply. The paper was reviewed once again to improve grammar.

  1. Line 43 use “structural”

Reply. Revised as suggested.

  1. Line 55 ad 56, “For present research, 4 brown macroalgae (Dictyota dichotoma, Cystoseira barbata, Eri- 55 caria amentacea, and Sargassum hornschuchii)”………… don’t they have author citations for each of the algae you studied?

Reply. There is no need to cite any paper regarding these algae because we mentioned only that they were collected within the scope of BioProCro project.

  1. Line 59 – 61, “ Higher number of brown algae were cho- sen, since it is often-recognized high content of phenolic compounds in brown macroalgae……………..” Construction of sentence is too poor. Correction is needed in the entire text

Reply. Revised as suggested.

  1. (GAE)/L , (QE)/L, CE/L ……… the units mentioned are wrong probably. In the Abstract different units are given………….. clarify. You must mention the units with respected to fresh weight / dry weight of algal samples

Reply. The units are correct. TPC is usually expressed as mg GAE (gallic acid equivalents) per g or L of sample, although other works reported in the literature can use quercetin equivalents (QE) or CE (catechin equivalents) as was commented in the introduction. In this work, TPC values are expressed in mg GAE/L and mg GAE/g (dry weight) as is commented in Section 3.4 (see Figure 1) to be able to compare the results obtained with those of the literature, since some authors refer the measurements to L and others to g.

Line 94 – 96, “Here, the efficiency and greenness of these two extraction techniques to obtain extracts rich in bioactive compounds is compared, considering the role of the extraction solvent to modulate the phenolic profile…………… sentence construction is too bad.

Reply. Revised as suggested.

  1. Line 118, 50 W, what does W stand for?

Reply. This is unit for Ultrasound Power.

  1. TPC results in terms of mass/volume……. How have you determined? The SD values are too wide in ranges………….

Reply. TPC was determined employing the well-known Folin-Ciocalteu (FC) colorimetric method described by Singleton and Rossi in 1965. Nowadays it is the most employed methodology to measure the TPC. In this work, a modification of the Zhang's guidelines [50] for microtitration in 96-well plates was employed. The procedure is detailed in Section 3.4. It is important to keep in mind that TPC, as well as other spectrophotometric measurements, is an estimated index of the amount of polyphenols contained in the sample. For this reason, the SD values are so wide. To know the exact concentration of polyphenols, it is necessary to use advanced analysis techniques such as LC-MS/MS, also used in this work.

  1. No use of representing the same data in Table and Figure form. Omit any one. Representation of total phenol content in mg GAE/g of dry or fresh weight extract is needed only.

Reply. Thanks for the suggestion. In the revised version, the data is only represented in Figure 1 (Table 1 was removed and the other tables were renumbered accordingly).

Line158, specie?

Reply. Revised as suggested.

  1. Principles? Why capital letter?

Reply. No need for capital letter. It is revised.

  1. Why the collected aliquots have been assayed several times? What is the use of that? Can’t those be pooled and then assayed? The units must be mentioned in mg GAE/ gram sample in both the extraction methods.

Reply. The main objective of the work was the identification and quantification of polyphenols different from the much more investigated phlorotannins, but which are known to contribute significantly to the bioactivities of the extracts. If the samples were pooled and analyzed, the concentration of polyphenols in the sample would be diluted since more than 70% of the polyphenols are detected in the first 5 mL as it is shown in Figure 4 . In this way, another objective of the work was achived: the use of a miniturized methodology with minimum solvent consumption to obtain extracts rich in bioactive compounds.

TPC units are already expressed as mg GAE/g for both techniques (see Table 1 in the revised version).

Change the figure and table likewise

Reply. As was previously commented, units for TPC values are already expressed as mg GAE/g in Figure 4a and in Table 1.

IC50 vlaues must be mention in mg or microgram/ml

Reply. The authors appreciate the suggestion, but for a direct comparison with the values reported by other authors, we prefer to keep the units in mg/L.

  1. Table 2 and Figure 5. Comparison of the (a) IC50, (b) TPC, (c) AA values for the first aliquot obtained by 253 MSPD and UAE. (a) IC50, of what? Of antioxidant activity using ABTS radical scavenging activity? Then what are the meanings of IC50 bargraph and ABTS ? How have you determined ABTS scavenging activity ?

Reply. The IC50 is the concentration of an antioxidant-containing substance required to scavenge 50% of the initial ABTS radicals. The lower the IC50 value, the more potent is the substance at scavenging ABTS and this implies a higher antioxidant activity. The procedure to evaluate the ABTS scavenging activity is detailed in Section 3.5. Now, in the revised version the calculations to obtain the IC50 values are included at the end of Section 3.5 for a better understanding..

  1. minority polyphe- nols? Use a better English word

Reply. Revised as suggested.

  1. Line 266, 267, Hence, the identification and characterization of other polyphenols in the target algae is scarce………… grammatical error

Reply. Revised as suggested.

  1. Table 3. 3-hydroxybenzoic acid + 4- hydroxybenzoic acid…………….. what is the significance of + sign? Were they not detected individually? What are their concentrations then? Did you quantify?

Reply. Both compounds are isomers with the same MS/MS transitions and very similar retention time. When the individual standards are injected, both compounds can be distinguished, but in real samples, it is very difficult to distinguish them. For that reason, the sum of concentration of both compounds is expressed. In this case, a calibration curve was perfomed taking into account the sum of both compounds (peak area) to quantify them.

  1. Line 299 Punctuation is missing

Reply. Revised as suggested.

  1. location of the -OH hydroxyl group? Write in a proper way

Reply. Revised as suggested.

  1. IN each case of UAE and MSPD, which solvent were used? I have found no comparison between EtOH-W and methanol extraction in each case of UAE and MSPD………………………????

Reply. The solvents were corrected. For UAE, both methanol and ethanol/water were employed whereas, based on these preliminary results, only ethanol/water was employed as solvent in MSPD.

  1. Line 359 was placed was placed…. Remove one

Reply. Revised as suggested.

  1. Line 371 “20 and 160 mg/L” Express in mg or micro/ml or microlitre in the entire text

Reply. As was previously commented, for comaprative prurposes, these units were employed.

  1. Line 377, In ABTS assay why have you used DPPH reagent?

Reply. It was a mistake. The correct reagent is ABTS.

  1. Line 407, greenness point of view, use proper englis

Reply. Revised as suggested.

Reviewer 3 Report

The study presented here is interesting and shows the identification of phenolic compounds from 4 brown macroalgae and one red macroalga collected in the Adriatic Sea. A focus on only brown macroalgae would be more relevant, as I do not understand why authors added a red macroalga… as this red macroalga seems not interesting (low phenolic content). The study is interesting as the authors used green processes and LC-MS separation/detection/identification of phenolic compounds. I wonder about the final goal of this study, as I do not find any originality in the study conducted.

After reading the manuscript, I however recommend this manuscript for a potential publication in the Journal Marine Drugs. But before that, I propose a major revision of the manuscript before it could be accepted for publication.  I explain the various points for improvement below.

Title of the manuscript: as a revision based on molecular works, highlighted a polyphyly of the genus Cystoseira which was then revised. The taxonomic accepted name of Cystoseira barbata is now Gongolaria barbata

Why do authors include one species of red algae among 4 species of brown macroalgae?

If the objective is screening then there are not enough species and one would expect several species of red algae and also several species of green algae.

In the present study, authors used 4 species of brown seaweeds and on species of red seaweed. Both groups do not produce same metabolites and then I do not understand the logic to use both in this work. under the objective of the manuscript.

Abstract: remove the information about mixture ethanol/water as it is not tested in the study presented here

Paragraph beginning at line l48: Authors develop metabolites co-extracted with phenolic compounds

Please replace polysaccharides by carbohydrates as not only polysaccharides are co-extracted with phlorotannins. There is also mannitol… which is not a polysaccharide…

In the same section (line 53-54) authors develop the different bonds between phloroglucinol units (C-C and diaryl ether). Some species produced fucophlorethols (especially Sargassaceae species) and then this kind of species could produce C-C and diaryl ether bonds

Then please add or/and between each kind of bonds

Some reviews exist on phlorotannins in Sargassaceae and it will be interesting to add this kind of review

When authors develop studies on phenolic content from Cystoseira species, it will be interesting to read all studies, as unless I am mistaken, some relevant reviews are absent from this manuscript.

Figure 1. I am surprise to read phenolic content in gallic acid equivalent

Phlorotannins are polymers of phloroglucinol and not polymers of gallic acid. Why authors used gallic acid as standard?

It will be more interesting to have the content in phloroglucinol equivalent to be able to compare with international studies…

Concerning solvents used in the study (Methanol and a mixture of Ethanol:Water), I do not understand the logic: why using a toxic solvent (Methanol) and a hydrethanolic mixture (not pure Ethanol)?

Table 1. Please note the real name of species of the genus Sargassum as it is noted Sargassum sp. In the Table.

Unless I am mistaken, Figure 1 and Table present similar results.

Figure 2. Please improve the legend of the figure as, at the moment, the figure is not comprehensive without the text

Table 2. I do not understand the unit used for TPC (/g) and TPC (/L)

Per gram of what? g of extract? g of algae? Dry weight? Wet weight?

Figure 5. Please note the test behind the IC50 as any reader doesn’t understand what is presented on the Figure 5. Please increase the size of characters as it is not well lisible

I am surprise to see no positive controls associated to the activity test. The positive control is crucial to perform any test.

Table 3. I am surprise about the detection of flavonoids in Brown algae and the absence of detection of phloroglucinol on Ericaria amentacea, as phloroglucinol was detected in another species of Ericaria, E. tamariscifloia in Jégou et al. (2015).

Figure 6. Please replace ametacea by the good name of species of your specimen: amentacea

Also, use the good name of genus for the species C. barbata => Gongolaria then write G. barbata instead of C. barbata

Line L320: Please note Gongolaria instead of Cystoseira

I really do not understand why there is a red macroalga within studied specimens, as the majority of macroalgae are brown seaweeds

Line l359: removed the repetition “was placed”

Section 3.4 (lines L363 -373

I do not understand why authors used gallic acid to estimate the phenolic content, as it is not the monomer at the origin of phlorotannins in brown seaweeds

It is problematic as it do not reflect the reality and authors won’t be able to compare with the international literature as it is a convention to use the monomer of compounds tested, and then to use phloroglucinol   

Section 3.5 (lines 374-382)

I do not understand the tests carried out, as in this section Materials & Methods, authors explained the ABTS test… and in the results section, we have results about the ABTS test (AA) and a graph about IC50 which is not explained… Which test is the IC50 for?

Line l410: I do not understand the beginning of the sentence: up to 15 flavonoids and non flavonoids were quantified

It would have been interesting to use activity tests for a field of application mentioned in the conclusion: anti-ageing activity test for the cosmetics field, ORAC test for the food field, etc

Because in this manuscript, the ABTS test is used + another test (IC50) but neither test has any industrial value.

Bibliographic references

Line l471: please put in italics Sargassum wightii

Lines l533: please correct authors as at the moment, the first names (anaelle, Leon, Laurent, etc) are given instead of the family names!

Author Response

Comments to the Author: The study presented here is interesting and shows the identification of phenolic compounds from 4 brown macroalgae and one red macroalga collected in the Adriatic Sea. A focus on only brown macroalgae would be more relevant, as I do not understand why authors added a red macroalga… as this red macroalga seems not interesting (low phenolic content). The study is interesting as the authors used green processes and LC-MS separation/detection/identification of phenolic compounds. I wonder about the final goal of this study, as I do not find any originality in the study conducted. After reading the manuscript, I however recommend this manuscript for a potential publication in the Journal Marine Drugs. But before that, I propose a major revision of the manuscript before it could be accepted for publication.  I explain the various points for improvement below.

Reply. Authors would like to thank you for the comments. We adopted the corrections and provided replies to your questions and remarks which we beleived that improved the quality of the manuscript.

  1. Title of the manuscript: as a revision based on molecular works, highlighted a polyphyly of the genus Cystoseira which was then revised. The taxonomic accepted name of Cystoseira barbata is now Gongolaria barbata

Reply. Thank you for your comment. We corrected the name to Gongolaria barbata.

  1. Why do authors include one species of red algae among 4 species of brown macroalgae?

Reply. Brown macroalgae are well known to contain phenolic compounds and therefore 4 species were chosen. One red macroalgae was chosen for comparison with brown macroalgae. Higher number of brown algae were chosen, since it is well known that brown macroalgae are rich in phenolics compared to red or green ones which is usually associated with phlorotannins.

  1. If the objective is screening then there are not enough species and one would expect several species of red algae and also several species of green algae.

Reply. We agree that for more detail comparison it would be interesting to use more species of red algae and more species of green macroalge. However, our research was focused on brow macroalgae while one red macroalgae was used for comparision. Higher number of brown algae were chosen, since it is well known that brown macroalgae are rich in phenolics compared to red or green ones which is usually associated with phlorotannins

  1. In the present study, authors used 4 species of brown seaweeds and on species of red seaweed. Both groups do not produce same metabolites and then I do not understand the logic to use both in this work. under the objective of the manuscript.

Reply. The focus on this research was on brown macroalgae since it is known that they contain phenolic compounds. Red macroalga was used for the comparison. Higher number of brown algae were chosen, since it is well known that brown macroalgae are rich in phenolics compared to red or green ones which is usually associated with phlorotannins.

  1. Abstract: remove the information about mixture ethanol/water as it is not tested in the study presented here.

Reply. In the preliminary experiments, both methanol and ethanol/water were employed. In view of the results, the mixture ethanol/water was employed as solvent to compare both extraction techniques (UAE and MSPD) for all target algae.  

  1. Paragraph beginning at line l48: Authors develop metabolites co-extracted with phenolic compounds. Please replace polysaccharides by carbohydrates as not only polysaccharides are co-extracted with phlorotannins. There is also mannitol… which is not a polysaccharide…

Reply. The sentence has been changed.

  1. In the same section (line 53-54) authors develop the different bonds between phloroglucinol units (C C and diaryl ether). Some species produced fucophlorethols (especially Sargassaceae species) and then this kind of species could produce C-C and diaryl ether bonds. Then please add or/and between each kind of bonds.

Reply. Thanks for the information. It has been included in the manuscript.

  1. Some reviews exist on phlorotannins in Sargassaceae and it will be interesting to add this kind of review. When authors develop studies on phenolic content from Cystoseira species, it will be interesting to read all studies, as unless I am mistaken, some relevant reviews are absent from this manuscript.

Reply. Several references have been included in the revised version.

  1. Figure 1. I am surprise to read phenolic content in gallic acid equivalent. Phlorotannins are polymers of phloroglucinol and not polymers of gallic acid. Why authors used gallic acid as standard? It will be more interesting to have the content in phloroglucinol equivalent to be able to compare with international studies…

Reply. As was included in Section 3.4, the TPC was determined employing the Folin-Ciocalteau method described by Singleton and Rossi [49] employing a modification of the Zhang's guidelines [50] for microtitration in 96-well plates. These methodologies use gallic acid as a reference standard and, although in this case the use of phloroglucinol would be most suitable, in our lab we have the methodology standardized employing gallic acid for all analysis. Besides, as is commented at the end of the introduction section, one of the objectives of the work is to explore minor polyphenols different from the much more investigated phlorotannins, but which are known to contribute significantly to the bioactivities of the extracts.

  1. Concerning solvents used in the study (Methanol and a mixture of Ethanol:Water), I do not understand the logic: why using a toxic solvent (Methanol) and a hydrethanolic mixture (not pure Ethanol)?

Reply. Methanol and other polar solvents such as ethanol or 1-propanol are listed as environmentally safe in the industrial SSGs (solvent selection guidelines) and they are at the top pf the list of green chemicals (E. Yilmaz et al. Chapter 5. Type of green solvents used in separation and preconcentration methods, in New generation green solvents for separation and preconcentration of organic and inorganic species, Elsevier, 2020, 207-266; JM. Kokosa, Selectin an extraction solvent for a greener liquid phase microextraction (LPME) mode-based analytical method, Trac-Trend Anal. Chem, 118 (2019) 238-247). Regarding the use of anf hydroorganoc mixture, most polyphenols are very polar compounds but other are less polar. For this reason, ethanol/water was employed to to extract as many compounds with different polarities as possible in a single step. 

  1. Table 1. Please note the real name of species of the genus Sargassum as it is noted Sargassum sp. In the Table. Unless I am mistaken, Figure 1 and Table present similar results.

Reply. Thanks for the comment. In the revised version, Table 1 was removed and the other tables were renumbered accordingly

  1. Figure 2. Please improve the legend of the figure as, at the moment, the figure is not comprehensive without the text.

Reply. The legend has been modified for a better understanding.  

  1. Table 2. I do not understand the unit used for TPC (/g) and TPC (/L). Per gram of what? g of extract? g of algae? Dry weight? Wet weight?

Reply. As is detailed in Section 3.4, the TPC units are expressed as mg of GAE (gallic acid equivalent) per L of extract or per gram of sample (algae) (dry weight). We consider both units to be able to compare with other works reported in the literature.

  1. Figure 5. Please note the test behind the IC50 as any reader doesn’t understand what is presented on the Figure 5. Please increase the size of characters as it is not well lisible

Reply. The quality of the figure has been improved.

  1. I am surprise to see no positive controls associated to the activity test. The positive control is crucial to perform any test.

Reply. We completely agree with you. It was not commented in the manuscript but, both positive and negative controls were performed. For AA, negative control was the ABTS solution (without sample), whereas the positive one was employed using trolox at a known concentration.

  1. Table 3. I am surprise about the detection of flavonoids in Brown algae and the absence of detection of phloroglucinol on Ericaria amentacea, as phloroglucinol was detected in another species of Ericaria, E. tamariscifloia in Jégou et al. (2015).

Reply. In this work 60 target polyphenols were analyzed (see Table S1) by LC-MS/MS but, phloroglucinol was not included in this list as the main objective of this work was to quantify minor polyphenols in the target species.

  1. Figure 6. Please replace ametacea by the good name of species of your specimen: amentacea

Reply. The name has been corrected.

  1. Also, use the good name of genus for the species C. barbata => Gongolaria then write G. barbata instead of C. barbata. Line L320: Please note Gongolaria instead of Cystoseira.

Reply. Thank you for your comment. We have changed the name as you suggested.

  1. I really do not understand why there is a red macroalga within studied specimens, as the majority of macroalgae are brown seaweeds

Reply. The focuses the research was on brown macroalgae since it is known that they contain phenolic compounds. Red macroalga was used the comparison. Higher number of brown algae were chosen, since it is well known that brown macroalgae are rich in phenolics compared to red or green ones which is usually associated with phlorotannins.

  1. Line l359: removed the repetition “was placed”

Reply. It was corrected

  1. Section 3.4 (lines L363 -373). I do not understand why authors used gallic acid to estimate the phenolic content, as it is not the monomer at the origin of phlorotannins in brown seaweeds. It is problematic as it do not reflect the reality and authors won’t be able to compare with the international literature as it is a convention to use the monomer of compounds tested, and then to use phloroglucinol   

Reply. As was commented before, TPC was determined by the Folin-Ciocalteau method described by Singleton and Rossi [49] employing a modification of the Zhang's guidelines [50] for microtitration in 96-well plates. These methodologies use gallic acid as a reference standard. Besides, as is commented at the end of the introduction section, one of the objectives of the work is to explore minor polyphenols different from phloroglucinol, which are known to contribute significantly to the bioactivities of the extracts.

  1. Section 3.5 (lines 374-382). I do not understand the tests carried out, as in this section Materials & Methods, authors explained the ABTS test… and in the results section, we have results about the ABTS test (AA) and a graph about IC50 which is not explained… Which test is the IC50 for?

Reply. Now, in the revised version the calculations to obtain the IC50 values are included at the end of Section 3.5 for a better understanding.

  1. Line l410: I do not understand the beginning of the sentence: up to 15 flavonoids and non flavonoids were quantified

Reply. As was previously commented, LC-MS/MS analysis working in SRM mode was performed to quantify 60 target compounds (see in Table S1). 15 of them were detected and quantified in the samples and, their concentration is shown in Table. The sentence has been modified for a better understanding. 

  1. It would have been interesting to use activity tests for a field of application mentioned in the conclusion: anti-ageing activity test for the cosmetics field, ORAC test for the food field, etc. cause in this manuscript, the ABTS test is used + another test (IC50) but neither test has any industrial value.

Reply. We completely agree with the reviewer. It would be very interesting to use the activity tests in applications but, this work is the first step to know the composition of the obtained extracts. Future research will be focused on real applications and, the corresponding tests with industrial value will be used.

  1. Bibliographic references. Line l471: please put in italics Sargassum wightii. Lines l533: please correct authors as at the moment, the first names (anaelle, Leon, Laurent, etc) are given instead of the family names!

Reply. The corrections have been made.

Round 2

Reviewer 3 Report

Authors replied to my questionings and corrections. Then, now I am ok with this version